# An Admittance Control Method Based on Parameters Fuzzification for Humanoid Steering Wheel Manipulation

**DOI:** 10.3390/biomimetics8060495

**Published:** 2023-10-19

**Authors:** Tuochang Wu, Junkai Ren, Chuang Cheng, Xun Liu, Hui Peng, Huimin Lu

**Affiliations:** 1College of Intelligence Science and Technology, National University of Defense Technology, 109 Deya Road, Kaifu District, Changsha 410073, China; wutuochang21@nudt.edu.cn (T.W.); chengchuang@nudt.edu.cn (C.C.); 2School of Mechanical Engineering, Shanghai Jiao Tong University, Shanghai 200240, China; liux_robot@sjtu.edu.cn; 3College of Computer Science and Technology, Central South University, Changsha 410017, China; hui-peng@csu.edu.cn

**Keywords:** humanoid manipulation, admittance control, parameter fuzzification, steering wheel manipulation

## Abstract

Developing a human bionic manipulator to achieve certain humanoid behavioral skills is a rising problem. Enabling robots to operate the steering wheel to drive the vehicle is a challenging task. To address the problem, this work designs a novel 7-DOF (degree of freedom) humanoid manipulator based on the arm structure of human bionics. The 3-2-2 structural arrangement of the motors and the structural modifications at the wrist allow the manipulator to act more similar to a man. Meanwhile, to manipulate the steering wheel stably and compliantly, an admittance control approach is firstly applied for this case. Considering that the system parameters vary in configuration, we further introduce parameter fuzzification for admittance control. Designed simulations were carried out in Coppeliasim to verify the proposed control approach. As the result shows, the improved method could realize compliant force control under extreme configurations. It demonstrates that the humanoid manipulator can twist the steering wheel stably even in extreme configurations. It is the first exploration to operate a steering wheel similar to a human with a manipulator by using admittance control.

## 1. Introduction

With the development of intelligent robots, there is an increasing demand for robots with various manipulation skills. Steering, a simple but commonly used human action, has attracted the attention of many researchers [1]. It aims to continuously rotate a rigid disc-like object, such as a wheel or valve, to a given configuration. Benefiting from the curious and exquisite physiological structure of the human arm, steering manipulation is an easy thing to do for humans [2]. Meanwhile, for manipulators, it is still a challenging problem to realize stable directional guidance when in contact with rigid disc-like objects [3].

After our verification, due to limitations in control efficiency and machine performance, robot humanoid driving has rarely been extensively studied in previous years. In 2015, Rojas et al. [4] proposed a three-step approach to achieve the wheel turning task in simulation. This method demonstrates high accuracy in controlling vehicle rotation angles, however, without considering dynamic constraints. Therefore, what about similar steering operations? For small-scale tuning objects, such as valves, Xing et al. [5] greatly simplify the steering manipulation problem and formulate it into a two-stage framework. It first grasps the valve with a special gripper tightly and then twists the valve by rotating only the end joint. Since only the end joint is free when twisting the valve, the workspace is very limited. In addition, the scale of their target objects also depends on the gripper, which is always small. Jiang et al. [6] propose a DDPG-based force-aware guidance strategy to screw a door handle with a UR humanoid manipulator. After training with many data, it can deal with the problem of changing force or torque when interacting with the door handle. Still, this method requires a time-consuming training process. Another similar piece of research is Latifinavid [7], who designed a 3-DOF Parallel Stabilizing Robot, which realizes stable control of the platform through kinematic modeling. All three need to face approximate steering problems, but the complexity is far from reaching steering wheel control. From above, we can see that the interaction between the manipulator and the environment cannot be separated from force control. Researchers use methods such as impedance control, admittance control, and fuzzy control to cope with interaction tasks. Yang et al. [8] proposed a whole-body impedance control approach for humanoid wheeled robots. This approach realizes safe interaction and compliant operation with unknown environments. Li et al. [9] proposed an NFMPC scheme for reliable tracking control by estimating the unknown physical interaction and external dynamics of the robot system. Some methods, such as admittance control, have been widely studied in industrial scenarios such as metal polishing [10], component assembly [11,12] and other dexterous manipulations [13,14]. Simply relying on trajectory tracking or servo control can hardly meet the full needs of efficient interaction between the manipulator and the steering wheel.

In this work, we introduce our own humanoid robotic arm design based on task requirements at the beginning. The key is to implement a lighter weight by imitating the structure of a human arm. Then, we firstly propose an admittance control method that achieves the stable steering wheel operation with a humanoid manipulator. The key first is to attempt to ensure stable interaction between the manipulator and the steering wheel. However, we found that the manipulator may often reach singularity and lose stability with the failure of the admittance controller at the near-limit operating distance. Therefore, a parameter fuzzification strategy is proposed to improve the control effect. Figure 1 shows the schematic diagram of the steering wheel with the humanoid manipulator. In addition, considering the traditional 6-DOF industry manipulators usually can not be equipped with the flexible operation function for imitating humans [15]. Thus, in this paper, a redundant 7-DOF manipulator is designed based on the bionic structure of the human arm.

A suitable humanoid robot platform is the first consideration for steering wheel control. Xin et al. [16] made a track-legged humanoid robot, called Dexbot, for dexterous manipulation. Sun et al. [17] made a humanoid dual-arm mobile robot, called BIT-DMR, for complex operations. Their single dexterous arm weighs 9.5 kg and is too long to be used in narrow driving spaces.

The contributions of this work are three-fold and can be summarized as follows:A 7-DOF humanoid manipulator is designed with a dexterous movement space. Imitating a human’s wrist, this provides a dexterous joint. Approaching human arm weight makes the manipulator lighter. Oblique installation offers a larger workspace. It is suitable as a research platform for the manipulator’s human-like operation.An admittance control approach is first applied to solve the safe interaction with the steering wheel. Different admittance controllers were applied to the end movement of the manipulator to achieve force position hybrid control during steering wheel operation. It can realize stable and compliant manipulation of the steering wheel with the 7-DOF humanoid manipulator.An admittance controller based on parameters fuzzification is proposed, which is designed to adapt different operating configurations. Three simulation environments are set up for small, medium, and large operating distances away from the steering wheel. By fuzzifying the inertia, damping, and stiffness parameters differently, our controller performs better than classic admittance controllers.

The rest of this paper is organized as follows. Section 2 formulates the problem of the steering wheel operation. Section 3 introduces the 7-DOF humanoid manipulator, its kinematics and dynamics, and the controller design. To verify the proposed approach, extensive experiments are performed in Section 4. Finally, conclusions can be found in Section 5. The implementation of our method in Coppeliasim is publicly available at our github homepage https://github.com/nubot-nudt/Steering_wheel_with_7Dof_arm_sim, last accessed on 17 August 2023.

## 2. Problem Formulation

The overall process of the manipulator operating the steering wheel is a complex and challenging set of tasks. The robot needs to transport to the front of the steering wheel, after which the manipulator trajectory planning is carried out according to the vision information processing to obtain the steering wheel size and attitude information. This research focuses on the project of how the humanoid manipulator grips the steering wheel and then steadily completes the action of rotating the steering wheel. Manipulating the steering wheel with a humanoid manipulator requires the following basic considerations for its execution.

Firstly, the task process of manipulating the steering wheel is a problem of maintaining trajectory tracking. The operating space of the steering wheel is usually a regular circular motion in the plane, and the end-effector needs to track this circular motion while performing self-rotation to keep the hand always facing the center of the movement during the operation of the steering wheel. Figure 2 shows the schematic diagram of the end-effector-operated steering wheel movement, where $\omega_{r}$ is the rotation speed of the steering wheel, and $\omega_{s}$ is the rotation speed of the end-effector. From the geometry, it can be obtained that the steering wheel rotation angle $\theta_{r}$ is equal to the end-effector rotation angle $\theta_{s}$ per unit of time, so $\omega_{r}=\omega_{s}$ describes the end motion constraint of the humanoid manipulator.

Secondly, cars will take turning moments between the tires and the ground when the vehicle turns. At the end of that turn, the suspension system will reposition the wheels and the body through its inherent reversion force so that the body will automatically return to its position. The EPS (Electric Power Steering) in most cars adaptiveiy adjusts the steering wheel deflection angle to increase handling performance [18]. While the humanoid manipulator is stiff in the position-tracking process, the vibration on the end-effector will damage the motor. Therefore, it is challenging to solve the problem of pulling the steering wheel with an end-effector and reducing the slewing vibration. It is necessary to design an admittance controller to realize the force control and active compliance of the end-effector [19].

Lastly, due to the impact of the design of the carrier platform, in some more extreme cases, the robot enters the vehicle with little space left for the arm to operate the steering wheel, and the manipulator may approach the singularity. At that time, the output of the admittance controller may drive the arm to break through the stability boundary and make the arm a singularity with the unstable steering wheel operation. Thus, an improvement strategy that makes the manipulator control method suitable for different manipulating spaces is worth considering.

In addition, the steering wheel manipulation task has vertical tilt, vibration interference, and manipulation rate in practical applications, which are not within the scope of the main study of this paper and are therefore not discussed further.

## 3. Methodology

### 3.1. Platform Design

In this paper, we adopt a self-designed 7-DOF humanoid manipulator concerning the arm of a male adult (1.7 m in height), shown in Figure 3. The robotic arm consists of seven motors, including seven rotational degrees of freedom. The shoulder joints (No.1–3) and elbow joint (No.4) use the planetary reduction joint actuator QDD-PR60-36 with a reduction ratio of 36. The wrist joints (No.5–7) use the planetary gear actuator QDD-NE30-36. The performance parameters of the motor are shown in Table 1. The torque and rotational speed are close to the daily activity data of the human arm [20,21]. The brachium and forearm are made of carbon fiber material with lengths of 274 mm and 265 mm. A full arm weight of 5 kg is close to the average human data. The Inspire-RHF-6-BXF dexterity hand is used to realize the gripping action of the steering wheel. It contains six DOF in whole, two degrees of freedom for the thumb, and one degree of freedom for each of the other four fingers.

To enlarge the operating space of the robot in front, the first joint is installed oblique 45 degrees, which makes the robot arm inverse solution more difficult but it can be solved by changing the definition of the base coordinate system. In addition, for the need to carry large loads, crossed roller bearings are used at the non-flanged output of the motor to improve the stiffness of the robotic arm. Moreover, joints 5, 6 and 7 are orthogonally arranged to imitate the human wrist and achieve some actions such as the ’folding wrist’ and ’buckling wrist’. Joint 5 is set forward to the elbow to improve wrist mobility, which is different from most 7-DOF robotic arms. It makes the end motion close to human and reduces the possibility of wrist singularity in the forward working space. By combining the movements of these seven joints, this manipulator can replicate the intricate and precise movements of a human arm.

### 3.2. Kinematic and Dynamic Modeling

As for kinematic solutions, modeling the arm’s structural body is required. Usually, the direct kinematic modeling process of the humanoid manipulator is to derive the end-effector in Cartesian space by calculating the rotation angle of each joint, while the inverse kinematics is to obtain the rotation angle of each joint by deducing the end-effector in Cartesian space in the reverse direction [22]. The arm configuration determines the relationship between the humanoid manipulator joint space and the end motion in Cartesian space and is represented by the D-H parameter [23]. The zero position state of the redundant manipulator and the corresponding D-H parameters are shown in Figure 4a and Table 2.

The inverse kinematic solution of the humanoid manipulator plays an important role in robot control. The Jacobi matrix represents the relationship between the joint rotation velocity and the end tool actuator motion velocity, while the Jacobian of a redundant manipulator is a non-square matrix, and its generalized inverse matrix is generally obtained by Singular Value Decomposition (SVD). The speed of N joints’ rotation is denoted as $\dot{q}=\left[\dot{\theta_{1}},\dot{\theta_{2}},\cdots ,\dot{\theta_{n}}\right]$, the 6 DOF motion of the end-effector is indicated as $\dot{p}=\left[\dot{x},\dot{y},\dot{z},\omega_{x},\omega_{y},\omega_{z}\right]$, and they are related by the Jacobi matrix, as expressed in the equation below
(1)$$\dot{p}=J\left(\theta \right)\dot{q}$$

The Jacobi matrix of the redundant manipulator is rectangular, represented by its pseudo-inverse as $\dot{p}=J^{+}\dot{q}$ [24]. Depending on the humanoid manipulator’s specific configuration, its Jacobian can be treated as a general linear equation system solution problem, and under the condition that the Jacobi matrix rows are of full rank, the solution is as follows
(2)$$\dot{q}=J^{+}\dot{p}+\left(I-J^{+}J\right)\dot{q_{0}}$$

Dynamics modeling is required because of the interaction between the humanoid manipulator and the steering wheel. The kinetic equations of the humanoid manipulator are derived from the Newton-Euler equation as follows, where M(q) is the mass matrix of the humanoid manipulator, $V(q,\dot{q})$ is the centrifugal force and Gauche force vector, G(q) is the gravity vector, and τ is the external output torque of the humanoid manipulator [25]
(3)$$M\left(q\right)\ddot{q}+V\left(q,\dot{q}\right)+G\left(q\right)=\tau$$

The relationship between the force *F* acting on the actuators at the end of the humanoid manipulator and the torque $\tau_{ext}$ applied to the joint can be related by the Jacobi matrix, such as $\tau_{ext}=J^{T}\left(q\right)F$, and the final dynamics modeling is expressed by the following equation
(4)$$M\left(q\right)\ddot{q}+V\left(q,\dot{q}\right)+G\left(q\right)=\tau -J^{T}\left(q\right)F$$

### 3.3. Admittance Controller Design Based on Parameters Fuzzification

This section focuses on the task of steering wheel operation by proposing an admittance controller based on parameters fuzzification, which consists of a trajectory tracking strategy to maintain circular motion through velocity control in Cartesian space [4]. An admittance controller supple strategy based on force sensors at the end-effector, and parameters fuzzification to improve the performance to adapt to different scenarios is used.

For the first question in the previous section, the end-effector is required to rotate while being able to accurately track a circular motion trajectory to always keep one axis pointing to the center of the circular motion, and this motion constraint $V_{goal}={\left[v_{x}^{B},v_{y}^{B},0,\omega_{x}^{E},0,0\right]}^{T}$ is described under Cartesian space as follows
(5)$$\left\{\begin{array}{c} \begin{array}{c} \omega_{x}^{E}=\omega_{desire}^{E}\\ v_{x}^{B}=\omega_{x}^{E}Rcos\left(\omega_{x}^{E}\Delta t\right)\\ v_{y}^{B}=\omega_{x}^{E}Rsin\left(\omega_{x}^{E}\Delta t\right) \end{array} \end{array}\right.$$
where $\omega_{x}^{E}$ is the angular velocity of the rotational motion of the end-effector in the x-axis direction under the end coordinate system E, $v_{x}^{B}$ and $v_{y}^{B}$ are the velocity of the translational motion of the end-effector in the x-axis and y-axis under the base coordinate system B, respectively, R is the radius of circular motion, and $\omega_{x}^{E}$ determines the rotational speed of the manipulating steering wheel. Since only the operation performed by the humanoid manipulator after gripping the edge of the steering wheel needs to be considered, this decision can be adapted to different types of humanoid manipulator end-effectors.

For the second question, it is necessary to design the admittance controller to realize the active compliance of the humanoid manipulator. Impedance/admittance control was first proposed by HOGAN [26] in 1985, and the goal of this method is to achieve the ideal dynamic relationship between the robot end position and end contact force, which is highly adaptable and can be applied to the design of robot motion control in both free space and constrained space. The following equation represents its second-order model
(6)$$\left\{\begin{array}{c} F_{adm}=M_{m}{\ddot{x}}_{e}+D_{m}{\dot{x}}_{e}+K_{m}x_{e}\\ x_{e}=x-x_{d} \end{array}\right.$$
where $M_{m}$ is the inertia coefficient matrix, $D_{m}$ is the damping coefficient matrix, and $K_{m}$ is the stiffness coefficient matrix. $F_{adm}$ is the 6-dimensional force generated by the admittance model to the environment, *x* and $x_{d}$ are the actual and desired poses, respectively [27]. As mentioned above, the end-effector needs to track a given circular trajectory, so the positional deviation $x_{e}^{\prime }$ generated by $F_{adm}$ alone is expressed as follows
(7)$$x_{e}^{\prime }=x-\left(x_{d}+{\dot{x}}_{d}\Delta t\right)$$

The velocity deviation ${\dot{x}}_{e}^{\prime }$ is obtained by differentiating $x_{e}^{\prime }$, and ${\dot{x}}_{e}^{\prime }$ is differentiated to obtain the acceleration deviation ${\ddot{x}}_{e}^{\prime }$. The admittance model is further rewritten as the following equation
(8)$$F_{adm}=M_{m}{\ddot{x}}_{e}^{\prime }+D_{m}{\dot{x}}_{e}^{\prime }+K_{m}x_{e}^{\prime }$$

Combining the external force $F_{ext}$ which acts on the actuator at the end of the humanoid manipulator and the output $F_{adm}$ of the admittance model, the final velocity control quantity $V={\left[v_{x},v_{y},v_{z},\omega_{x},\omega_{y},\omega_{z}\right]}^{T}$ in Cartesian space is given by the following equation
(9)$$\left\{\begin{array}{c} V=V_{goal}+\int_{t}^{t+1}\frac{\Delta F}{M_{m}}dt\\ \Delta F=F_{ext}-F_{adm} \end{array}\right.$$

The flow chart of the humanoid manipulator compliant control is shown in Figure 5.

To improve adaptability at different operating distances [28], the $M_{m}$, $D_{m}$ and $K_{m}$ parameters of the admittance controller are fuzzified. Fuzzy control is a rule-based control that directly adopts language-based control rules and establishes the mapping relationship between controlled quantities and semantic rules through the control experience of field operators or the knowledge of relevant experts [29]. Figure 6 shows several pictures of different distance limits of the humanoid manipulator body from the steering wheel, corresponding to the degree of joint4.

The construction of a Fuzzy Inference System (FIS) is based on the Takagi-Sugeno model [30]. The change of $\theta_{4}$ away from 90° and its rate of change are used as fuzzy control inputs, which are represented by X,Y respectively. Because of the task space being in front of the manipulator, only $\theta_{4}>90$ is considered. *X* contains membership functions as zero (Z), positive small (PS), positive medium (PM) and positive big (PB). *Y* contains membership functions as negative big (NB), negative small (NS), zero (Z), positive small (PS), and positive big (PB). The fuzzy control output includes parameters of the admittance controller, which contain membership functions as low (L), medium (M), and high (H). $M_{m},D_{m},K_{m}$ use the relevant fuzzified rules, which are shown in Table 3.

The farther away the manipulator is from the steering wheel, the closer it reaches the limit attitude of the manipulator, and the more likely it becomes a singularity. In this case, the system will be unstable if the admittance controller produces the output due to vibration or steering wheel slewing torque that pulls the end-effector to the limit state [10]. It can also be seen from Figure 6 that the angle of the humanoid manipulator elbow joint (motor 4) has a linear relationship with the operating distance, so the fuzzy rule can be set by this experience to adjust the corresponding inertia, damping, and stiffness parameters, with the initial posture elbow joint angle of 90° as the dividing line. Figure 7 shows the flow chart of our control method.
(10)$$\left\{\begin{array}{c} M_{m}=M_{0}+\Delta M_{m}\\ D_{m}=D_{0}+\Delta D_{m}\\ K_{m}=K_{0}+\Delta K_{m} \end{array}\right.$$
where $\Delta M_{m}$, $\Delta D_{m}$ and $\Delta K_{m}$ conform to the following rule, $\theta_{4}$ represents the angle of motor 4 at the initial attitude of the motion. Δθ is determined by the deviation of $\theta_{4}$ from the minimum value of the current interval at the initial attitude, such as $\Delta \theta =\theta_{4}-\theta_{min}$
(11)$$\Delta M_{m}=\left\{\begin{array}{cc} 0 & ,\theta_{4}\in \left[80,145\right]\\ 0.1*\Delta \theta & ,\theta_{4}\in \left(145,165\right] \end{array}\right.$$
(12)$$\Delta D_{m}=\left\{\begin{array}{cc} 0 & ,\theta_{4}\in \left[80,115\right]\\ 0.1*\Delta \theta & ,\theta_{4}\in \left(115,165\right] \end{array}\right.$$
(13)$$\Delta K_{m}=\left\{\begin{array}{cc} 10*\Delta \theta & ,\theta_{4}\in \left[80,115\right]\\ 50*\Delta \theta & ,\theta_{4}\in \left(115,145\right]\\ {\left(\Delta \theta \right)}^{3} & ,\theta_{4}\in \left(145,165\right] \end{array}\right.$$

## 4. Simulation and Analysis

The computer device used is a Lenovo Blade 7000 K with Inter core i7-11700F CPU, 32 GB of RAM, 1.5 TB of storage and NVIDIA RTX-3060Ti GPU. The simulation software used is Coppeliasim in version 4.1, which is also called Vrep in older versions. The physics engine of simulation is Bullet 2.78, and the simulation step is set to 10 ms. The operating system used is Ubuntu18.04 based on Linux for an easy coding and debugging. A Robot Operating System (ROS, Melodic) is used to communicate with the Coppeliasim and the ROS message release frequency is set to 100 Hz. The model is imported through an urdf file to ensure the accuracy, the humanoid manipulator is installed on the left side of the stand, and the steering wheel (radius is 0.23 m) is placed in front of the bracket, facing the operation direction of the humanoid manipulator. The end coordinate system and the base coordinate system are shown in Figure 8. The initial parameters of inertia, damping, and stiffness are set constant as follows
(14)$$\left\{\begin{array}{c} M_{0}=\left[1,1,1,1,1,1\right]\\ D_{0}=\left[0.1,0.1,0.1,5.5,5.5,5.5\right]\\ K_{0}=\left[500,500,500,500,500,500\right] \end{array}\right.$$

### 4.1. Verification of the Inverse Kinematic Solution

The simulation sets the initial angle $\theta_{4}$ to 90°. The manipulator is set to start from the grip point, then make a clockwise rotation of 90 degrees. It will immediately return back to the origin after the reverse movement along the trajectory, then stop simulation. As is shown in Figure 9, the maximum error of the x-axis in the trip is 0.029 m and the maximum error of the y-axis is 0.017 m, as shown from the results of the motion control effect in line with expectations. The force at the end of the free-motion humanoid manipulator shown in the bottom-right figure is around 0.3 N. No contact with the wheel occurs in the first 5 s, which does not produce the actual control effect. Furthermore, each axis receives unexpected force that causes a jump in direction if it takes none-force control.

On the one hand these data prove the correctness of the kinematic inverse solution, on the other hand they show that the task cannot be accomplished using motion control alone. In comparison, the left figure in Figure 10 shows that the manipulator performed good interaction with the steering wheel with an admittance controller.

### 4.2. Verification of the Admittance Controller Based on Parameters Fuzzification

According to the range of parameter fuzzification, we set the initial angle $\theta_{4}$ of motor 4 to 90°, 130°, and 155°, respectively, to simulate the three conditions of the humanoid manipulator operating the steering wheel at a moderate, large and extreme distance, where the humanoid manipulator starts from the gripping point. It rotates 90° clockwise and then immediately stops the movement. Then the x-axis desire force is set to 1.5 N, simulating the humanoid manipulator to apply direct pressure on the steering wheel to maintain stable contact.

Firstly, we started the simulation with an initial condition of small operating distances. As is shown in Figure 10, the result of the admittance controller is 0.140 N in standard deviation(Std) of the x-axis, which is smaller than that of our controller result (0.179 N). It represents that the vibration of the contact force with the admittance controller is smaller. However, there is a jump in the rotation of the steering wheel to some specific angles (about 60°), which is due to the lack of stiffness in the direction of rotation. In contrast, the admittance controller with parameters fuzzification is applied with force in the x-axis with a mean of 1.552 N. Meanwhile, the former was 1.571 N. Compared with the expected value of 1.5 N, the performance of the two controllers is very close in this condition.

Then, we started the simulation with an initial condition of medium operating distances. As is shown in Figure 11, the mean of our controller output is 1.592 N, and the max force is 1.779 N on the x-axis. The admittance controller is 1.853 N and 2.183 N, respectively, which give a worse performance than the former. The results show the former’s advantages with a small error to pre-set force of 1.5 N. Compared with Figure 10, the admittance controller with parameters fuzzification performs better as the initial $\theta_{4}$ angle increases.

Finally, we started the simulation with an initial condition of large operating distances. As is shown in Figure 12 and Figure 13, pictures are cut with desired steering wheel rotation angles of 0°, 30°, 50°, 70°, 90°. The former, Figure 13a, after a short period of stable force application, whose x-axis force suddenly increased beyond the limit reached 13.49 N, makes the arm become a singularity. Meanwhile, the latter, Figure 13b, can apply a stable force control in the range of 2 N to 4.4 N, and complete the task. The results show that the method with our controller can complete the steering wheel operation in near-limit operating distance, which can hardly be finished by an admittance controller only.

With all the data above organized, a comparative table has been completed as Table 4. When the operating distance gradually increases, the performance of the admittance controller with parameters fuzzification becomes increasingly superior to a single admittance controller.

## 5. Conclusions

This paper proposes an admittance control method with parameters fuzzification to achieve compliant steering wheel operation with a self-designed humanoid manipulator. Firstly, the challenges of operating the steering wheel by a humanoid manipulator are analyzed, including the problem of unstable interaction and the problem of operation at a near-limit distance. Secondly, the kinematic and dynamic modeling of the self-designed humanoid manipulator is introduced. Furthermore, a kind of admittance controller is designed, whose inertia, damping, and stiffness coefficients are fuzzified according to the operating distance of the steering wheel. After that, simulation results provide intuitive data to verify the effectiveness and reliability of this control method, which can complete the steering wheel operation at a near-limit operating distance.

The biomimetic design of redundant manipulators will contribute to achieving flexibility similar to human arms. Much biology research on human body structure, such as muscle group movement and scapulohumeral rhythm, can help design high-performance robotic arms. Meanwhile, studying the humanoid skills of robots can help understand human action mechanisms and behavioral patterns. However, learning humanoid driving skills is a challenging task which requires domain knowledge in many engineering fields, such as mechanical design, adaptive control, multi-sensor fusion, and trajectory tracking, etc. The experiment results demonstrate the effectiveness of admittance control during the steering wheel operation with a redundant manipulator. As far as we know, this is the first publicly available exploration for robots to operate a steering wheel like humans with the help of control theory.

It is noteworthy that the proposed method is based on the ground-truth state as the input signal. However, it is difficult to measure the state precisely during the interaction between the steering wheel and the manipulator. The initial parameters need to be adjusted appropriately to achieve good control effect. What is more, our method relies on linearization functions with parameters fuzzification, which may be difficult to adapt to complex unknown environments. In future work, the development of humanoid skills in robots will be an important research direction. We will demonstrate more convincing experiments on our designed real platform. Adding damping systems is an efficient approach to improve the performance of the current controller. Latifinavid [7] designed a 3-DOF Parallel Stabilizing Robot and manufactured it to actively isolate the host vehicle’s disturbing motions. Currently, the use of only force sensors restricts the robot to a limited perception scenario. Therefore, by adding cameras, radar, and other sensors, we can increase the robot’s perception capabilities [31]. By integrating neural networks, reinforcement learning, and other methods, it may even be possible to achieve autonomous driving [9].

## Figures and Tables

**Figure 1 biomimetics-08-00495-f001:**
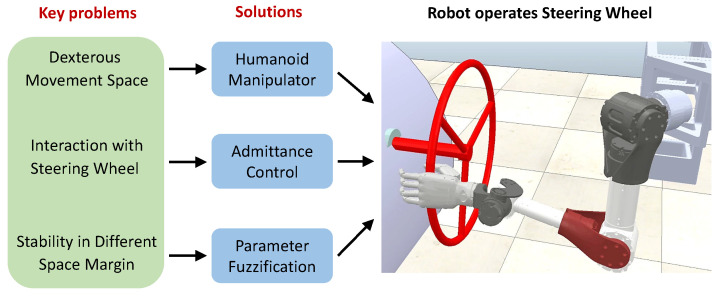
Structure of the steering wheel with a humanoid manipulator.

**Figure 2 biomimetics-08-00495-f002:**
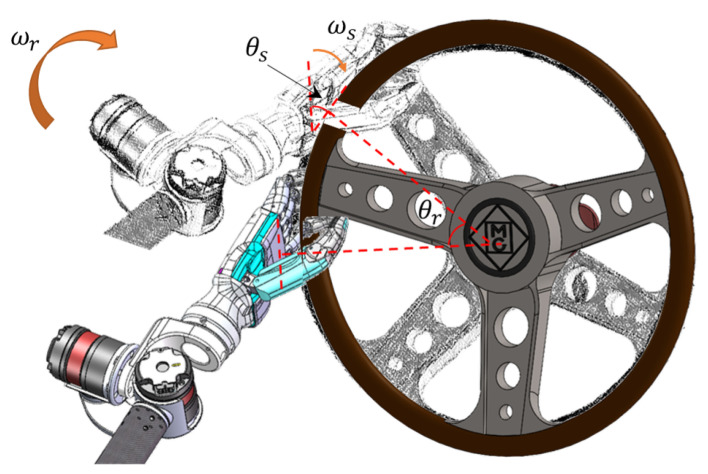
Steering wheel operation process.

**Figure 3 biomimetics-08-00495-f003:**
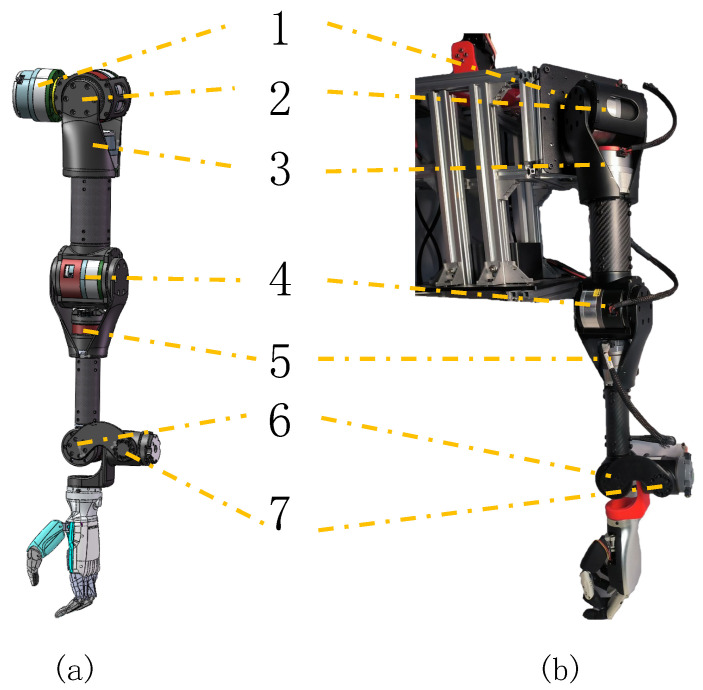
Our designed 7-DOF humanoid manipulator. (**a**) 3D-model; (**b**) real platform.

**Figure 4 biomimetics-08-00495-f004:**
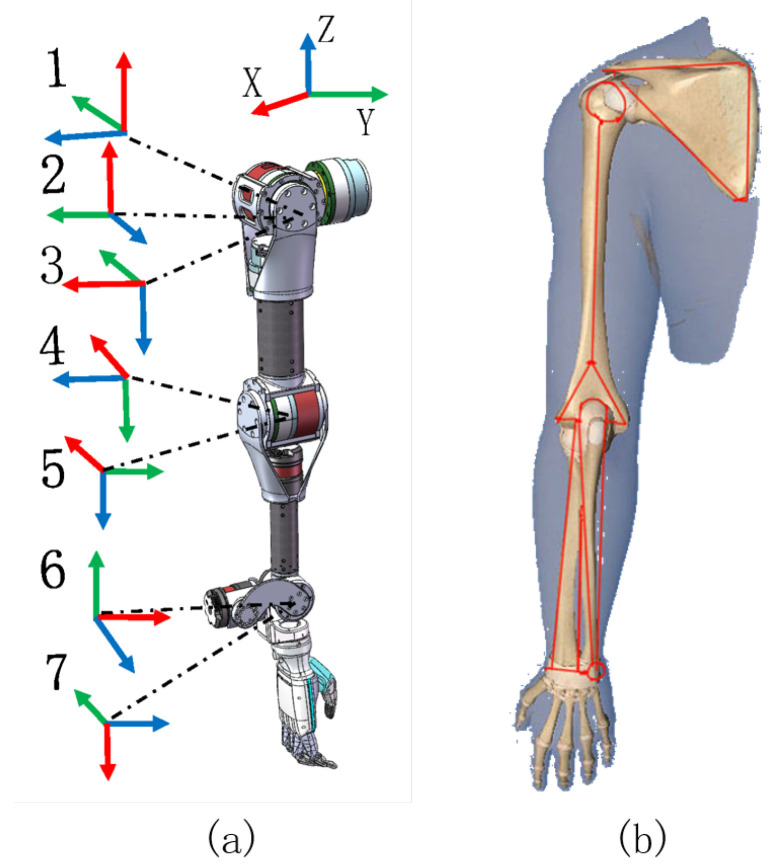
(**a**) The zero position state of the designed humanoid manipulator. (**b**) skeleton structure of human arm. The red, green, and blue lines represent the force value in x-axis, y-axis, and z-axis respectively.

**Figure 5 biomimetics-08-00495-f005:**
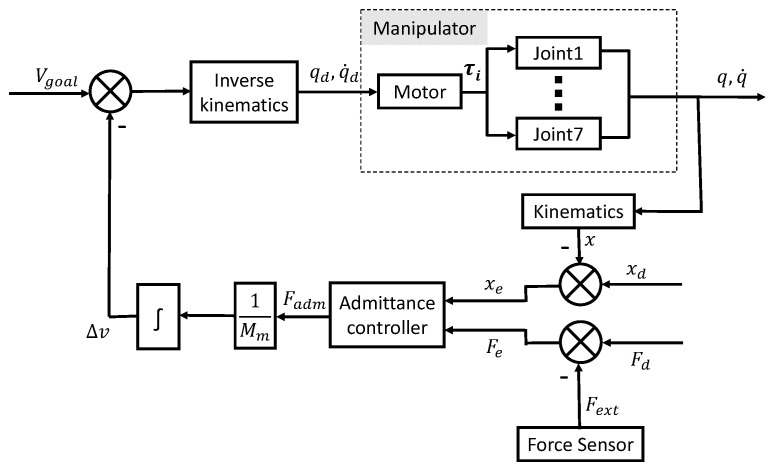
Humanoid manipulator admittance control.

**Figure 6 biomimetics-08-00495-f006:**
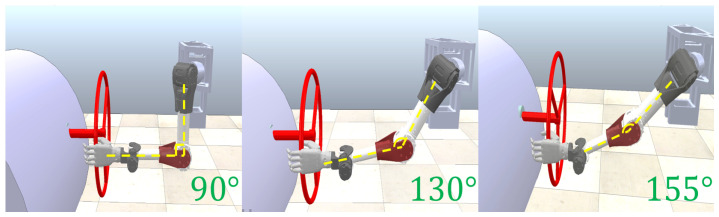
The start position of the humanoid manipulator at different operating distances.

**Figure 7 biomimetics-08-00495-f007:**
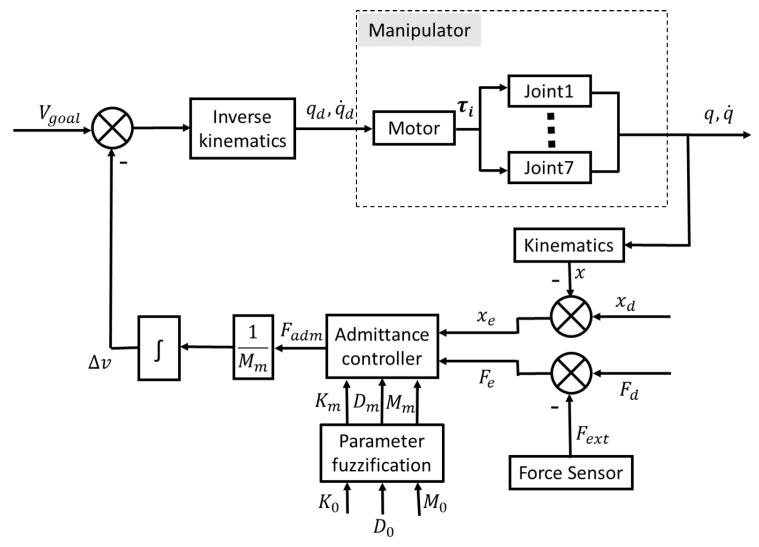
Humanoid manipulator admittance control based on parameters fuzzification.

**Figure 8 biomimetics-08-00495-f008:**
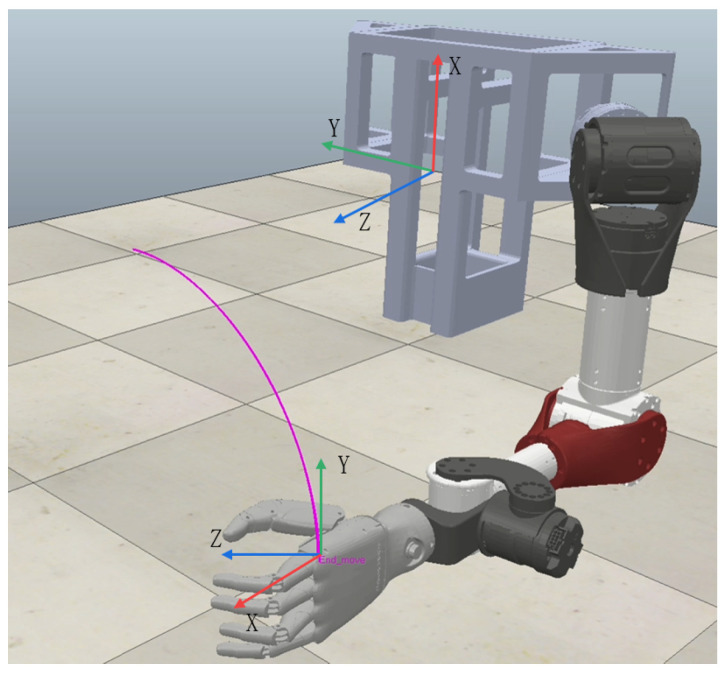
Trajectory tracking process and definitions of coordinate system.

**Figure 9 biomimetics-08-00495-f009:**
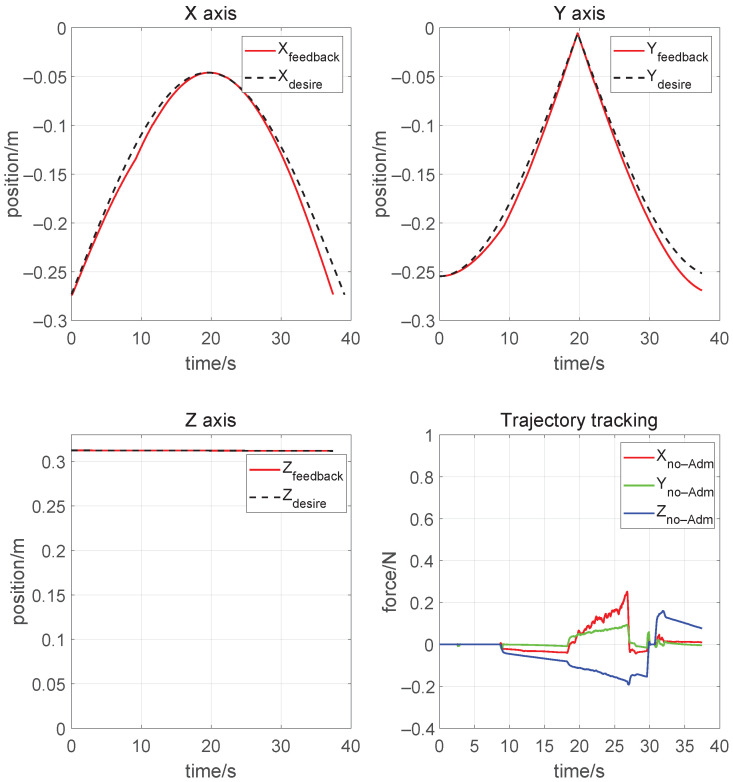
Results of trajectory tracking. The red, green, and blue lines represent the force value in x-axis, y-axis, and z-axis respectively.

**Figure 10 biomimetics-08-00495-f010:**
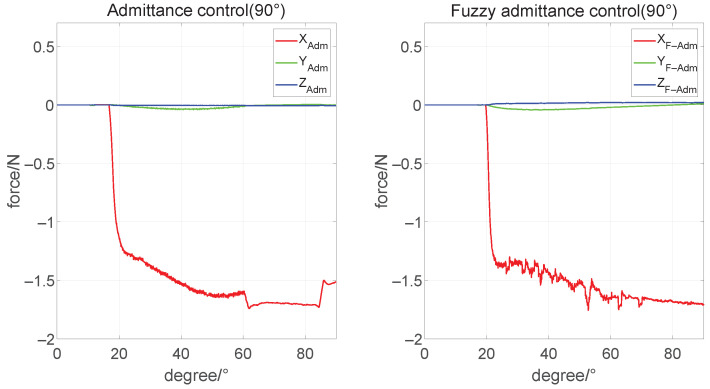
Force tracking under $\theta_{4}=90{}^{\circ}$ condition.

**Figure 11 biomimetics-08-00495-f011:**
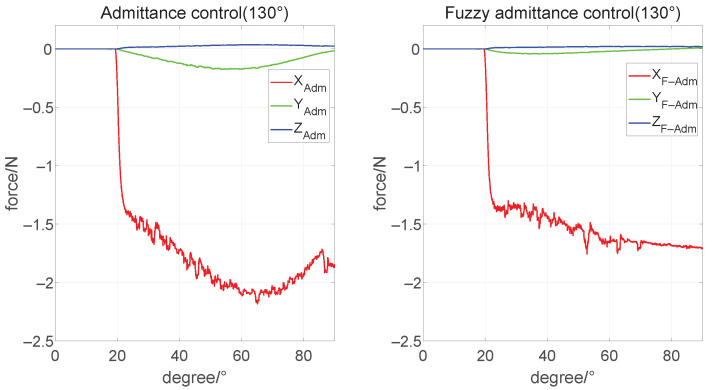
Force tracking under $\theta_{4}=130{}^{\circ}$ condition.

**Figure 12 biomimetics-08-00495-f012:**
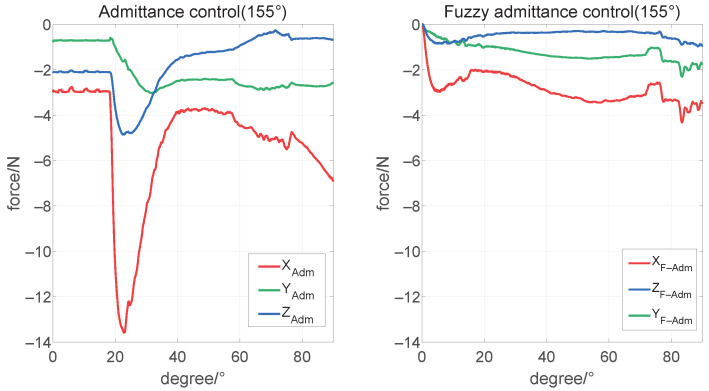
Force tracking under $\theta_{4}=155{}^{\circ}$ condition.

**Figure 13 biomimetics-08-00495-f013:**
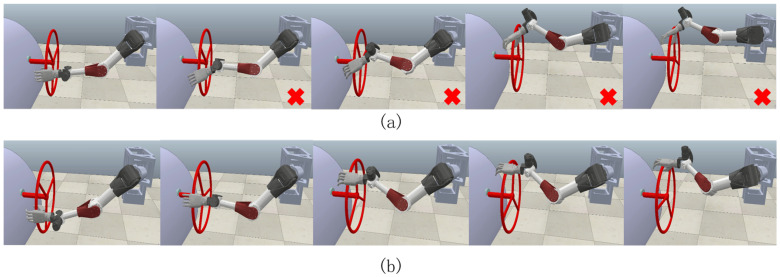
Operation steering wheel movement under $\theta_{4}=155{}^{\circ}$ condition. (**a**) The admittance controller (fail); (**b**) the admittance controller based on parameters fuzzification (success).

**Table 1 biomimetics-08-00495-t001:** Performance parameters of the motor.

Motor	Reduction Ratio	Torque (max)/Nm	Speed (max)/rpm	Weight/kg
1	36	23 (69)	83 (117)	0.72
2	36	23 (69)	83 (117)	0.72
3	36	23 (69)	83 (117)	0.72
4	36	23 (69)	83 (117)	0.72
5	36	6.5 (19)	111 (167)	0.3
6	36	6.5 (19)	111 (167)	0.3
7	36	6.5 (19)	111 (167)	0.3

**Table 2 biomimetics-08-00495-t002:** DH parameters.

Joint	$\alpha_{i}{/}^{\circ }$	$a_{i}$/mm	$d_{i}$/mm	$\theta_{i}{/}^{\circ }$
1	90	0	0	$\theta_{1}$
2	−90	0	0	$\theta_{2}+90{}^{\circ}$
3	90	0	$d_{3}\left(274\right)$	$\theta_{3}+90{}^{\circ}$
4	−90	0	0	$\theta_{4}$
5	−90	0	$d_{5}\left(265\right)$	$\theta_{5}+90{}^{\circ}$
6	−90	0	0	$\theta_{6}-90{}^{\circ}$
7	0	$a_{7}$	0	$\theta_{7}$

**Table 3 biomimetics-08-00495-t003:** Parameters fuzzified rules of *M_m_*, *D_m_*.

		Y	NB	NS	Z	PS	PB
	*M_m_*						
X							
Z			L	L	L	L	L
PS			L	L	L	L	L
PM			L	L	L	M	M
PB			L	L	M	M	M
		**Y**	**NB**	**NS**	**Z**	**PS**	**PB**
	* **D_m_** *						
**X**							
Z			L	L	L	M	M
PS			L	L	L	M	H
PM			L	L	M	M	H
PB			L	M	M	H	H
		**Y**	**NB**	**NS**	**Z**	**PS**	**PB**
	* **K_m_** *						
**X**							
Z			L	L	M	M	H
PS			L	M	M	H	H
PM			M	M	H	H	H
PB			M	H	H	H	H

**Table 4 biomimetics-08-00495-t004:** Comparison of x-axis force under different initial $\theta_{4}$.

Method	Adm-Controller			Our-Controller		
Initial $\theta_{4}=$	90°	130°	155°	90°	130°	155°
Std/N	0.140	0.210	2.466	0.179	0.127	0.517
Mean/N	1.571	1.853	5.755	1.552	1.592	2.907
Maximum/N	1.740	2.183	13.490	1.756	1.779	4.329

NOTE: absolute values have been taken and calculations are based on data within steady state. Adm is admittance for short.

## Data Availability

All data generated or analyzed during this study are included in this paper or are available from the corresponding authors on reasonable request.

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
