# Peer review of "An Admittance Control Method Based on Parameters Fuzzification for Humanoid Steering Wheel Manipulation"

_biomimetics, 2023, doi:10.3390/biomimetics8060495_

Round 1
Reviewer 1 Report
Please open the attached file

Author Response
Dear Reviewer:
We are most grateful for your detailed review and constructive comments concerning our manuscript “A Fuzzy Admittance Control Method for Humanoid Steering Wheel Manipulation” (ID: biomimetics-2584135). The comments are valuable for revising and improving our paper and will be constructive guidance for our future work. We have modified the title into “An Admittance Control Method Based on Parameters Fuzzification for Humanoid Steering Wheel Manipulation”. Thank you for your kind suggestions.
Please see the attachment for details.

Reviewer 2 Report
The literature review in the Introduction is quite good, but in my opinion, the authors could have devoted more attention to the issue of admittance control for the field of robotics. It is not at all clear from the overview whether such a method of control was used for this area as well.
The authors state that the contribution of the paper is the proposal of a fuzzy admittance controller. In my opinion, it is not a fuzzy controller, but an admittance controller with fuzzification of its parameters. In terms of this comment, I do not agree with the title of the paper, and this fact should also be taken into account in the text of the entire paper.
The paper overall look is satisfying. The analysis of the issue is systematic, but a more detailed description of the fuzzification of the parameters of the admittance controller and also of the parameters used in the simulations is missing above all.
It is stated in the abstract that „extensive experiments were carried out“, but this is not obvious from the results presented in Chap. 4.1. Figure 3 shows the real platform of the 7DOF human manipulator, but the results are verified only by simulations.
The conclusion lacks a more detailed evaluation of the achieved results, primarily from the the novelty of the proposed solutions. It would be appropriate to compare the results with the previous method stated in the literature in order to demonstrate the superiority and the effectiveness of the proposed approach.
Comments:
If any, indicate restrictions for use proposed control strategy.
Do not use references to literature in the Abstract.
Figure 1, not Fig.1, is used correctly at the beginning of the sentence (p2/r49). The same applies to Fig.2 (p3/r83) and Fig.6 (p8/r.202).
In Fig.3, the individual joints (1-7) are not marked with numbers as stated in the text of the paper (p4/r112-113).
Do not use a dot at the end of a sentence followed by an equation (throughout the paper).
The methodology of parameter fuzzification is insufficiently described. It would be appropriate to state, for example, the rules for the fuzzy system of setting parameters, the type of fuzzy system, etc.
The specific parameters used for the simulation are not mentioned in the paper.
The English looks good, there are minor grammatical errors and incorrect wording in the article. It should be correctly “parameters are fuzzified” and not “parameters are fuzzed”.
I recommend language proofreading.
Author Response
Dear Reviewer:
We are most grateful for your detailed review and constructive comments concerning our manuscript “A Fuzzy Admittance Control Method for Humanoid Steering Wheel Manipulation” (ID: biomimetics-2584135). The comments are valuable for revising and improving our paper and will be constructive guidance for our future work. We have studied comments carefully and invested efforts to implement them.
We have modified the title into “An Admittance Control Method Based on Parameters Fuzzification for Humanoid Steering Wheel Manipulation”,accordingly. Thank you for your kind suggestion.
Please see the attachment for more details.

Reviewer 3 Report
Dear Researchers
Thanks for the valuable manuscript. There are some points that need to be addressed to improve the quality. Please find my comments as follows:
-
The theoretical contributions should be stressed in detail in the Introduction.
-
Please check all notations and equations carefully. Moreover, the use of English should be improved.
-
The advantages of the proposed algorithm over well-known algorithms should be stressed. I suggest mentioning/comparing the simulations with the results of the recent valid references.
-
In introduction, it is not enough to state the current work. It should be expanded and reconstructed. Including the motivation, the main difficulties, the main work, and the improvements compared with previous related works should be emphasised in this section.
-
The state of the art is not complete, and I suggest reviewing the following works as: potential resources to be reviewed in the introduction section of various techniques.
- https://doi.org/10.1016/j.neucom.2020.05.091
- https://doi.org/10.3390/fi15020084
- DOI: 10.1109/LRA.2022.3151401
- https://doi.org/10.1007/s10846-022-01795-x
-
The importance of the problem considered in this paper should be further addressed.
-
The types of software employed for solving the problem and for simulation experiments should be stated clearly.
-
The directions to further and improve the work should be expanded by adding a more detailed future recommendation section after the conclusions section.
Author Response
Dear Reviewer:
We are most grateful for your detailed review and constructive comments concerning our manuscript “A Fuzzy Admittance Control Method for Humanoid Steering Wheel Manipulation” (ID: biomimetics-2584135). As you are concerned, there are several problems that need to be addressed. The comments are valuable for revising and improving our paper and will be constructive guidance for our future work.
Besides, we have modified the title into “An Admittance Control Method Based on Parameters Fuzzification for Humanoid Steering Wheel Manipulation”. Thank you for your kind suggestions.
Please see the attachment for more details.

Round 2
Reviewer 2 Report
I am pleased with the revision of the original paper. I appreciate that the paper has been significantly improved according to my comments, including its title.
Author Response
Dear Reviewer:
We are most grateful for your detailed review and constructive comments concerning our manuscript “A Fuzzy Admittance Control Method for Humanoid Steering Wheel Manipulation” (ID: biomimetics-2584135). The comments are valuable for revising and improving our paper and will be constructive guidance for our future work. We have studied comments carefully and invested efforts to implement them.
Reviewer 3 Report
Dear Authors
Thanks for the revised version. all points have been addressed satisfactory.
Author Response
Dear Reviewer:
We are most grateful for your detailed review and constructive comments concerning our manuscript “A Fuzzy Admittance Control Method for Humanoid Steering Wheel Manipulation” (ID: biomimetics-2584135).
As you are concerned, there are several problems that need to be addressed. According to your nice suggestions, we have made up for many shortcomings. Thanks a lot!